# Rapid Identification of Tanshinone IIA Metabolites in an Amyloid-β_1-42_ Induced Alzherimer’s Disease Rat Model using UHPLC-Q-Exactive Qrbitrap Mass Spectrometry

**DOI:** 10.3390/molecules24142584

**Published:** 2019-07-16

**Authors:** Shuang Liang, Zijian Wang, Jiaqi Yuan, Jing Zhang, Xueling Dai, Fei Qin, Jiayu Zhang, Yaxuan Sun

**Affiliations:** 1College of Biochemical Engineering, Beijing Union University, Beijing 100191, China; 2Beijing University of Chinese Medicine, Beijing Research Institute of Chinese Medicine, Beijing 100191, China; 3Beijing Key Laboratory of Bioactive Substances and Functional Foods, Beijing Union University, Beijing 100191, China; 4School of Pharmacy, Binzhou Medical University, Yantai 264003, China

**Keywords:** Tanshinone IIA, Alzheimer’s disease, UHPLC-Q-Exactive Qrbitrap mass spectrometry, metabolic pathway, hippocampus, rat model, Morris water maze test, passive avoidance test

## Abstract

Alzheimer’s disease (AD) is a neurodegenerative disorder that damages health and welfare of the elderly, and there has been no effective therapy for AD until now. It has been proved that tanshinone IIA (tan IIA) could alleviate pathological symptoms of AD via improving non-amyloidogenic cleavage of amyloid precursor protein, decreasing the accumulations of p-tau and amyloid-β_1–42_ (Aβ_1–42_), and so forth. However, the further biochemical mechanisms of tan IIA are not clear. The experiment was undertaken to explore metabolites of tan IIA in AD rats induced by microinjecting Aβ_1-42_ in the CA1 region of hippocampus. AD rats were orally administrated with tan IIA at 100 mg/kg weight, and plasma, urine, faeces, kidney, liver and brain were then collected for metabolites analysis by UHPLC-Q-Exactive Qrbitrap mass spectrometry. Consequently, a total of 37 metabolites were positively or putatively identified on the basis of mass fragmentation behavior, accurate mass measurements and retention times. As a result, methylation, hydroxylation, dehydration, decarbonylation, reduction reaction, glucuronidation, glycine linking and their composite reactions were characterized to illuminate metabolic pathways of tan IIA in vivo. Several metabolites presented differences in the distribution of tan IIA between the sham control and the AD model group. Overall, these results provided valuable references for research on metabolites of tan IIA in vivo and its probable active structure for exerting neuroprotection.

## 1. Introduction

Alzheimer’s disease (AD) is an overwhelming neurodegenerative disorder that weakens the mental, memorial and cognitional ability of patients, eventually leading to death due to infectious diseases and pneumonia [1]. Neuropathological features of AD include the presence of extracellular amyloid plaques and intracellular neurofibrillary tangles [2]. The essential pathological hypotheses of AD contain acetylcholine deficiency, increased inflammatory response, β-amyloid (Aβ) and tau accumulation, and so forth [3,4,5]. Meanwhile, there have been increasing evidence suggesting the existence of various pathophysiological pathways, including brain metabolic dysfunction [6], axonal disintegration [7], synaptic degeneration and dysfunction [8], and cell membrane dysregulation [9]. There is plenty of research regarding alleviating AD via decreasing Aβ generation and accelerating its clearance, stabilizing tau and inhibiting its aggregation, and regulating apolipoprotein-E (ApoE) function [10]. The effects of some biologicals, such as β-site amyloid precursor protein-cleaving enzyme 1 (BACE1) inhibitors which are based on Aβ accumulation hypotheses, to alleviate AD have been explored [11]. In addition, numerous neuropeptides, such as orexin, substance P, neurotensin, and ghrelin, exert significant neuroprotection through preventing Aβ accumulation, inhibiting endoplasmic reticulum stress, and so forth [12]. Although research on AD has been comprehensively studied, there is still no effective therapy to slow the progression or cure it [13]. In consequence, it is indispensable to identify innovative active substances and explore its mechanism to relieve or prevent AD.

Danshen, also called as *Salvia miltiorrhiza*, or Chinese sage, has been universally used in treating various diseases, such as cardiovascular and cerebrovascular disease. Tanshinones are the main liposoluble components of Danshen and include several most significant bioactive constituents, such as tanshinone I (tan I), tanshinone IIA (tan IIA, Figure 1), and cryptotanshinone (CPT) [14]. Tan IIA has been reported to possess a large range of bioactivities, such as antioxidants [15,16], antitumor [17,18], anti-inflammatory [19,20], and so forth. In particular, it has been proved that tan IIA exerts significant neuroprotective effects via various mechanisms. Tan IIA could prevent human neuroblastoma cell line SH-SY5Y cells from glutamate-induced cytotoxicity via alleviating oxidative stress and reducing apoptosis [21]. Moreover, tan IIA could suppress the formation of unseeded amyloid fibril and disaggregate Aβ, which is the main constituent of extracellular amyloid plaques of AD patients [22]. However, tan IIA was able to reduce the accumulation of Aβ_1–42_ by accelerating non-amyloidogenic process, which might be regulated via the phosphatidylinositol 3-kinase/Akt pathway [23]. Furthermore, tan IIA was capable of ameliorating impaired learning and memory ability, caused by Aβ_1–42_ infusion, with the mechanism involving glycogen synthase kinase-3β (GSK-3β) and signal-regulated protein kinase (ERK) [24]. These clues indicate that tan IIA could exert crucial effects on the therapy of AD. Li et al. incubated tan IIA with rat liver microsomes in a water bath at 37 °C for 3 h, centrifuged the mixture, and acquired residues for metabolites analysis. In total, three hydroxyl metabolites of tan IIA were identified, including przewaquinone A, hydroxytanshinone and tanshinone IIB [25]. Sun et al. analyzed seven active components of tanshinone, which included tan IIA, via a sensitive HPLC/MS^n^ method. Dehydrotanshinone IIA, and tanshinaldehyde were identified, which were produced through dehydrogenation and oxidation of tan IIA respectively [26]. Wei et al. [27] also researched the metabolic pathway of tan IIA, but no novel metabolites were identified. Further, detections of metabolites of tan IIA, especially in an AD model, could be indispensable and helpful to reveal the action mechanism of tan IIA on AD.

Q-Exactive is an advanced hybrid Orbitrap mass spectrometer, which combines the high selectivity of quadrupole with excellent resolving power performance of Qrbitrap. The resolving power of Q-Exactive can reach to 140,000 FWHM at *m*/*z* 400 and the mass accuracy of Q-Exactive can reach less than 2 ppm mass error. The application of high–resolution full scan mode and MS/MS mode of ion fragmentation can achieve the high selectivity and sensitivity [28]. In total, Q-Exactive Orbitrap mass spectrometry is a qualitied technique for ion fragmentation scanning with high sensitivity, resolving power and mass accuracy [29] and has been comprehensively used in these fields such as metabolomics, proteomics, lipidomics, and so forth [30,31,32].

This study, firstly applied an UHPLC-Q-Exactive Qrbitrap mass spectrometry-based method to explore tan IIA metabolites in an AD rat model, which was significant to understand the underlying mechanisms of neuroprotective effect of tan IIA against AD.

## 2. Results

### 2.1. Effect of Aβ_1–42_ on Morris Water Maze Test of AD Rat Model

The Morris water maze (MWM) test was applied to examine learning and memory capability of experimental rats. Escape latency and numbers of target crossing were analyzed respectively. As shown in Figure 2A, during the training phase, the escape latency of both the sham control and model groups decreased as time increased. Compared to the sham control group, the rats of model group had significantly longer escape latency (*p* < 0.01). In a spatial probe test, compared to the sham control groups, Aβ_1–42_-induced rats had markedly longer escape latency (*p* < 0.01, Figure 2B) and fewer times in achieving the target (*p* < 0.01, Figure 2C). Thus, it indicated the learning and memory impairments in the model group.

### 2.2. Effect of Aβ_1–42_ on Passive Avoidance Test AD Rat Model

A passive avoidance test was used to further detect rat spatial memory abilities. As showed in Figure 2D, compared to the sham group, enterance latency of the model group was markedly lowered (*p* < 0.01), which further improved that memory ability was impaired by Aβ_1–42_ microinjection.

### 2.3. Effect of Aβ_1–42_ on Hippocampal Neuronal Structure of AD Rat Model

Nissl body is one of the characteristic structures of a neuron, and could get larger during the neurons’ synthesis of protein powerfully. Meanwhile, decreasing or disappearing of the Nissl body indicated that neurons were damaged. Therefore, Nissl staining could manifest cell structure and damage neurons in the hippocampus. As showed in Figure 3A, compared to those in the model group, the neurons in CA1 region of the hippocampus of the sham control group were arranged much closer and had higher distributional densities. As showed in Figure 3B, compared to the sham control group, the intensity of the Nissl staining of model group was lower, which indicated that the model group showed heavier neuronal loss.

### 2.4. Mass Fragmentation Behavior Analyses of Tanshinone IIA

For the purpose of obtaining a comprehensive fragmentation behavior of tan IIA, the standard solution of tan IIA was analyzed via UHPLC-Q-Exactive Orbitrap MS, which was helpful for tan IIA metabolites identification. In the positive ion mode, tan IIA showed [M + H]^+^ ion at *m*/*z* 295.1327 (C_19_H_19_O_3_, −0.15 ppm) with the retention time of 15.23 min. Several characteristic ions at *m/z* 293.1164 (C_19_H_17_O_3_, −0.82 ppm), *m*/*z* 281.1163 (C_18_H_17_O_3_, −0.92 ppm), *m*/*z* 277.1218 (C_19_H_17_O_2_, −0.51 ppm), *m/z* 267.1378 (C_18_H_19_O_2_, −0.15 ppm), *m*/*z* 267.1010 (C_17_H_15_O_3_, −0.57 ppm) and *m*/*z* 249.1269 (C18H17O, −0.49 ppm) were respectively produced by losing 2H, CH_2_, H_2_O, CO, 2CH_2_ and CO + H_2_O from [M + H]^+^ ion [22]. In addition, the other characteristic fragment ion, such as *m/z* 265.1216 (C_18_H_17_O_2_, 2.66 ppm), has been identified and was illustrated in Figure 4. 

### 2.5. Identification of Metabolites of Tanshinone IIA in Rats

Total ion chromatograms of urine, feces, heart, liver, kidney, brain and plasma samples, which were collected from experimental rats after oral administration of tan IIA, were analyzed by Thermo Xcalibur 2.1 workstation. Comparing the samples of the sham control and model groups with those of the control group, 37 metabolites were obtained in a positive ion mode. The proposed metabolic patterns of tan IIA are shown in Figure 5 and the corresponding data about metabolites are summarized in Table 1.

**M0**, **M11** and **M22**, which respectively possessed their [M + H]^+^ ions at *m*/*z* 329.1381, 329.1385 and 329.1379 (C_19_H_21_O_5_, mass error with −0.79, 0.61 and −1.52 ppm) with the retention time of 4.25, 7.45 and 9.61 min, were 34 Da more than tan IIA. The characteristic fragment ions at *m*/*z* 311, 283 and 265 were produced by successive losses of H_2_O, CO and H_2_O, which were coincident with the fragmentation behavior of tan IIA. Therefore, it was provisionally presumed that **M0**, **M11** and **M22** were produced via changing olefin of tan IIA into dihydrodiol.

Metabolites **M1** and **M8**, which showed protonated ions at *m*/*z* 325.1068 and 325.1072 (C_19_H_17_O_5_, mass error with −0.74 and 0.58 ppm) with the retention times of 4.36 and 6.6 min, were 30 Da higher than tan IIA and 2 Da less than double hydroxylation products of tan IIA like **M10**. The characteristic ions at *m*/*z* 307 [M + H − H_2_O]^+^ and *m*/*z* 279 [M + H − H_2_O-CO]^+^ of **M1** and at *m*/*z* 281 [M + H – OH − CO]^+^ of **M9** were consistent with cracking characteristics of tan IIA, which provide reliable reference for metabolites identification. Thus, **M1** and **M8** were tentatively identified as dihydroxylation and dehydrogenation products of tan IIA.

**M2** and **M7** respectively produced their [M + H]^+^ ions at *m*/*z* 281.1170 and 281.1172 (C_18_H_17_O_3_, error with −0.68 and −0.22 ppm) with the retention times of 4.75 and 6.22 min, which were deductively assigned as demethylation products of tan IIA because their protonated ions were 14 Da less than tan IIA. Moreover, both **M2** and **M7** yielded the characteristic fragment ion at *m*/*z* 263 [M + H − H_2_O]^+^ via losing H_2_O, which also confirmed the authors’ initial speculation.

**M3** and **M24** showed their protonated molecular ions at *m*/*z* 309.1121 and 309.1116 (C_19_H_17_O_4_, mass error with −3.41 and −3.51 ppm) with the retention times of 4.81 and 9.76 min. They were 14 Da more than tan IIA and 2 Da less than **M16**, **M18** and **M28**, which were hydroxylation products of tan IIA. As a result, **M3** was tentatively identified as hydroxylation and dehydrogenation product of tan IIA. However, the fragment ion of **M3** at *m*/*z* 267 [M + H − CO − CH_2_]^+^ was in accordance with the fragment pathway of tan IIA. **M24** were speculatively identified as isomers of **M5** in the present condition.

Metabolites **M4**, **M5**, **M25**, **M33** and **M35**, which individually possessed [M + H]^+^ ions at *m*/*z* 313.1438, 313.1437, 313.1435, 313.1434 and 313.1432 (C_19_H_21_O_4_, error with 1.26, 0.78, 0.27, −0.02 and −0.69 ppm) with the retention times of 5.71, 5.72, 9.87, 11.93 and 13.09 min. They were 18 Da more than tanIIA and a battery of fragment ions at *m*/*z* 295 [M + H − H_2_O]^+^, *m*/*z* 277 [M + H − 2H_2_O]^+^ and *m*/*z* 267 [M + H − H_2_O − CO]^+^ were produced, which were consistent with fragmentation behavior of tan IIA. Therefore, they were provisionally characterized as intramolecular hydrolysis products of tan IIA.

Metabolites **M6**, **M13** and **M27**, which showed their protonated ions at *m*/*z* 327.1226, 327.1223 and 327.1225 (C_19_H_19_O_5_, error with −0.06, −2.78 and −0.52 ppm) with retention times of 6.2, 8.7 and 10.3 min, were 32 Da higher than tan IIA. In addition, fragment ions at *m*/*z* 309 [M + H − H_2_O]^+^ and *m*/*z* 281 [M + H − 2H_2_O]^+^ were explored in their MS/MS spectra, which provided available evidence for our conjecture. Due to their retention times, **M6**, **M13** and **M27** were respectively identified as double hydroxylation products of tan IIA.

In ESI-MSspectra, metabolites **M10**, **M19** and **M30** provided their protonated molecular ions at *m*/*z* 311.1277, 311.1275 and 311.1280 (C_19_H_19_O_4_, error within −0.40, −0.92 and 0.56 ppm) with the retention times of 6.85, 9.27 and 11.28 min, respectively. They were 16 Da more than that of tan IIA, meaning that they might be hydroxylation products of tan IIA. In their MS/MS spectra, a battery of fragment ions at *m*/*z* 293 [M + H − H_2_O]^+^, *m*/*z* 283 [M + H − CO]^+^ and *m*/*z* 265 [M + H − CH_2_O]^+^ were identified. Based on the related references and retention times of these three metabolites, **M10**, **M19** and **M30** were respectively identified as Tanshinone IIB, 3α-Hydroxytanshinone IIA and Przewaquinone A [27].

**M9** and **M26**, which were eluted at 6.84 and 10.28 min, gave rise to their protonated molecular ions at *m*/*z* 265.1224 and 265.1222 (C_18_H_17_O_2_, error within 0.39 and −0.21 ppm). They were 28 Da less than **M10**, **M19** and **M30**, suggesting that **M9** and **M26** might be their dehydration and decarbonylation products. The characteristic fragment ion at *m*/*z* 247 [M + H − H_2_O]^+^ was explored in **M9**. Due to the elution orders in the reverse ODS column, **M9** and **M26** were respectively identified as the dehydration and decarbonylation products of **M19** and **M30**, which were hydroxylation products of tanIIA.

Metabolite **M12**, which produced [M + H]^+^ ion at *m*/*z* 291.1014 (C_19_H_5_O_3_, error with −2.54 ppm) with the retention time of 8.33 min, was preliminarily thought as a double dehydrogenation product of tan IIA since its protonated ion was 4 Da lower than tan IIA. In its ESI-MS/MS spectra, the characteristic fragment ion at *m*/*z* 273 [M + H − H_2_O]^+^ was explored, which was in accordance with the mass fragmentation behavior of tan IIA.

**M14** and **M37** gave rise to protonated ions at *m*/*z* 313.1062 and 313.1072 (C_18_H_17_O_5_, error with 1.48 and 0.32 ppm), which were eluted at 8.75 and 13.68 min, respectively. They were 18 Da more than tan IIA and 16 Da higher than **M17**, **M32** and **M36**. In ESI-MS/MS spectra of **M37**, the characteristic ions at *m*/*z* 295.0972 and *m*/*z* 267.1007 were yielded by successive dehydration. In addition, fragment ions at *m/z* 285 [M + H − CO]^+^ and *m*/*z* 269 [M + H − CO − OH]^+^ proved the existence of hydroxylation. Due to their retention times, **M37** was identified as a demethylation and dihydroxylation product of tan IIA and **M14** was tentatively assigned as an isomer of **M37**.

In ESI-MS spectra, **M15** provided protonated ion at *m*/*z* 297.1482 (C_19_H_21_O_3_, error with −1.11 ppm) with the retention time of 9.01 min, which were 2 Da lower than tan IIA. The fragment ions at *m*/*z* 279 and *m*/*z* 251 were produced via successively losing H_2_O and CO, which were consistent with the fragmentation pattern of tan IIA. Thus, it is provisionally interpreted that **M15** was produced by changing carbonyl of tan IIA to oxhydryl.

**M16**, **M18** and **M28**, which showed their respective [M+H]+ ions at *m*/*z* 293.1169, 293.1169 and 293.1168 (C_19_H_17_O_3_, error with −1.16, −2.83 and −3.24 ppm) with the retention times of 9.13, 9.27 and 10.62 min, were plausibly assigned dehydrogenation products of tan IIA since their deprotonated ions were upon the loss of 2 Da. The characteristic product ions at *m*/*z* 275 [M + H − H_2_O]^+^ were detected in their ESI-MS/MS spectra. In addition, the product ions of **M16** and **M28** at *m*/*z* 247 [M + H − H_2_O − CO]^+^ provided substantial evidence for identifying metabolites. Due to the literature reports, **M16** was identified as 1,3-dehydrotanshinone IIA, **M18** and **M28** was characterized as 1,2-dehydrotanshinone IIA or 2,3-dehydrotanshinone IIA [23].

Metabolites **M17**, **M32** and **M36**, whose elution times were 9.14, 11.58 and 13.53 min, possessed [M + H]^+^ ions at *m*/*z* 297.1123, 297.1126 and 297.1119 (C_18_H_17_O_4_, error with 0.688, 1.395 and −2.504 ppm) respectively. In their MS/MS spectra, characteristic ions at *m*/*z* 251 [M + H − CO − H_2_O]^+^ of **M17**, at *m*/*z* 279 [M + H − H_2_O]^+^ of M32 and at *m*/*z* 269 [M + H − CO]^+^ of **M36** were detected, which were in accordance with the fragment pattern of tan IIA. Based on their protonated ions and retention times, **M17**, **M32** and **M36** were respectively assigned as demethylation products of **M10**, **M19** and **M30**, which were putatively assigned as Tanshinone IIB, 3α-Hydroxytanshinone IIA and Przewaquinone A.

**M20** produced a protonated ion at *m*/*z* 471.1641(C_25_H_27_O_9_, error with −1.84 ppm) with the retention time of 9.46 min, which was 176 Da more than tan IIA. In ESI-MS/MS spectra, the characteristic ion at *m*/*z* 295 was obtained via losing a glucuronide moiety (176 Da), which provide reliable evidence for identifying metabolites. Moreover, the fragment ion at *m*/*z* 277 [M + H − GluA − H_2_O]^+^ was consistent with the fragment pathway of tan IIA. Therefore, **M20** was putatively thought of as tan IIA -*O*-glucuronide.

**M21** was eluted at 9.6 min and possessed [M + H]^+^ ion at *m*/*z* 267.1375 (C_18_H_19_O_2_, error with −1.78 ppm), which was 28 Da less than tan IIA. In its ESI-MS/MS spectra, the fragment ions at *m*/*z* 225 [M + H − CO − CH_3_]^+^and *m*/*z* 211 [M + H − CO − 2CH_3_]^+^ provided substantial evidence for identifying metabolites. Therefore, **M21** was putatively thought as a decarbonylation product of tan IIA.

**M23** and **M31**, 48 Da higher than tan IIA, produced their [M + H]^+^ ions at *m*/*z* 343.1175 and 343.1176 (C_19_H_19_O_6_, error with −0.45 and −0.01 ppm) with the retention times of 9.61 and 11.48 min. In their MS/MS spectra, fragment ions at *m*/*z* 325 [M + H − H_2_O]^+^, *m*/*z* 297 [M + H − H_2_O − CO]^+^, *m*/*z* 283 [M + H − H_2_O – CO − CH_3_]^+^ and *m*/*z* 279 [M + H − 2H_2_O − CO]^+^ were produced which accord with the fragmentation pattern of tan IIA. Therefore, it is tentatively identified that **M23** and **M31** were isomers of each other and were triple hydroxylation products of tan IIA.

Metabolite **M29** was eluted at 10.7 min and showed protonated ion at *m*/*z* 352.1541(C_21_H_22_O_4_N, mass error with −0.64 ppm), which was 57 Da higher than tan IIA. The fragment ion at *m*/*z* 310 [M + H − H_2_O − CO]^+^ was in accordance with the fragment pattern of tan IIA. Thus, it was putatively identified that **M29** was produced via combing tan IIA with a glycine.

**M34** gave rise to its [M + H]^+^ ion at *m*/*z* 325.1437 (C_20_H_21_O_4_, error with 0.66 ppm) with the retention time of 12.37 min and was 30 Da more than tan IIA. In ESI-MS/MS spectra, fragment ions at *m*/*z* 279 and 265 were yielded by the successive loss of H_2_O, CO and CH_3_, which were in accordance with the fragmentation pattern of tan IIA. Therefore, **M34** was speculatively assigned as methylation and hydroxylation product of tan IIA 

## 3. Discussion

In 1981, the Morris water maze (MWM) test was designed by Morris to investigate the learning and memory capability of experimental rats [33]. Meanwhile, the test has become one of the most universal methods for behavioral neuroscience [34]. The test was applied in our experiment to test spatial learning and memory impairment of AD model rats. In the place navigation test, the experiment rats had three trails per day for 4 days successively. Jiang et al. [35] trailed mice eight times in a navigation test, which was heavy stress for experimental animals. Body weight might be decreased sharply which could influence the conduction of the passive avoidance test. Meilin et al. [36] trailed rats twice per day during the place navigation test. Compared to the design of Meilin et al. [36], our experiment could compare the differences of spatial learning and memory between the sham control group and the model group more effectively.

Metabolism of tan IIA has been analyzed by Wei, Sun and Wu et al. [26,27,37] by HPLC/MS^n^, while the UHPLC/MS^n^ was utilized in our study to achieve superior resolution, sensitivity and speed. In addition, using the UHPLC-Q-Exactive Oribitrap mass spectrometry, some novel metabolites were putatively identified in our study, such as methylation(**M34**), dehydration(**M21**), decarbonylation (**M9**, **M26**), reduction reaction(**M12**, **M16**, **M18**, **M28**), glucuronidation(**M20**), glycine linking (**M29**), and so forth.

Our study analyzed metabolites of tan IIA in an Aβ_1–42_ induced AD model rats for the first time. Comparing the metabolic results of the AD model group with that of the sham control group, it was concluded that there were significant distributional differences between them. Metabolites **M2**, **M3**, **M10**, **M23**, and **M31** were only detected in the urine samples of the sham control group. Metabolites **M4** and **M5** were identified as intramolecular hydrolysis products of tan IIA, but they had differences in distributions. **M4** was detected in urine samples of the AD model group, while **M5** was recognized in the plasma sample of the sham control. Metabolites **M15**, **M17**, **M24**, **M26**, **M30**, and **M34** were only identified in the feces samples of the AD model rats. Moreover, **M29**, which formed by combing tan IIA with a glycine, was acquired in the urine, feces and kidney samples of the sham control groups. **M12** was only detected in brain samples of the sham control group. This could be a clue that tan IIA is able to go through the blood-brain barrier which suppresses effects exerting on AD inhibitors [24]. In addition, **M20**, which was putatively identified as tan IIA-*O*-glucuronide, was only detected in urine samples of AD model rats. However, **M27**, which was sensitively assigned as double hydroxylation products of tan IIA, was only found in feces samples of AD model rats. Therefore, **M20** and **M27** might be possible active structures of tan IIA to protect neurons and ease pathological features. Further research can focus on the exact structures identifying neuronal protections of **M20** and **M27**.

## 4. Materials and Methods

### 4.1. Chemicals and Reagents

Tan IIA was purchased from Beijing Spectrum Analysis Technology Co., Ltd. (Beijing, China). Aβ_1–42_ was commercially provided by GL Biochem Ltd. (Shanghai, China). Nissl staining solution was supplied by Beyotime Biotechnology Co., Ltd. (Shanghai, China). HPLC grade methanol, formic acid (FA), acetonitrile was purchased from Beijing RUIZHX Tech Co., Ltd. (Beijing, China). ProElutTM C18 solid phase extraction (SPE) cartridges (500 mg / 3 mL, 50 μm, 60 Å) were obtained from Dikma Technologies Co., Ltd. (Foothills Ranch, California, USA). All other usual chemicals and reagents used in the whole progress of the experiment were of analytical grade.

### 4.2. Preparation of Aggregated Soluble Aβ_1–42_ Oligomers

Aggregated soluble Aβ_1–42_ oligomers were prepared according to published methods [38,39,40]. Lyophilized Aβ_1–42_ peptide, placed in room temperature for 30 min, was dissolved in 100% hexafluroisopropanol (HFIP, Sigma, St. Louis, MI, USA) with a final concentration of 1 mM and then the solution was aliquoted in aseptic microcentrifuge tubes. HFIP was eliminated in vacuum conditions by a Speed Vac followed by storing the desiccated peptide film at −80 °C. In order to obtain aggregated Aβ_1–42_, the peptide film was re-dissolved in dimethyl sulfoxide (DMSO, Sigma, St. Louis, Missouri, USA) to 5 mM and then the final concentration of Aβ_1-42_ oligomers was adjusted to 2.5 μg/μL via adding phosphate-buffered saline (PBS, pH 7.4). Finally, the mixture was incubated at 4 °C for 24 h.

### 4.3. Animals and Drug Administration

Twenty seven male Sprague-Dawley rats (200–220 g) were commercially supplied by Beijing Vital River Laboratory Animal Technology (Beijing, China) and kept in a constant environment with temperature (22 ± 1 °C), humidity (65–75%) and a 12 h light/dark cycle. The experimental rats were allowed to access food and water ad libitum and acclimatized to conditions mentioned above for a week. All rats were divided randomly into three groups: Control (for blank biological samples, *n* = 7), sham control (*n* = 10) and model groups (*n* = 10). This experiment was carried out according to Figure 6. All experimental procedures and animal facilities were approved by the institutional Animal Care and Use Committee of the Center of Functional Inspection of Health Food, College of Applied Science and Humanities, Beijing Union University. The animal experiment, which was approved by above committee from December 2017 to June 2018 and numbered as 2017-59, had the same title as this paper. There were no significant differences in body weights and food intakes between the sham control group and the model group throughout the experiment.

### 4.4. Microinjection and Surgery

The intrahippocampal injection method was utilized to build the AD rat model. The rats were anesthetized by sodium pentobarbital (70 mg/kg, i.p.) and attached to a stereotaxic apparatus (RWD Life Science Co., Ltd., Shenzhen, China). A small incision was made to expose the bregma which was adjusted to the same horizontal plane with lambda [41]. Bilateral burr holes were drilled based on Paxinos and Watson rat brain atlas [42], and the stereotaxic coordinates were as follows: anteroposterior: −3.3 mm from bregma, medial/ventral: ±0.17 mm, dorsal/ventral: −3.6 mm. All rats in the model group were injected bilaterally with 5 μL aggregated Aβ_1–42_ oligomers, while rats in the sham group were injected bilaterally with 5 μL sterile PBS. The injection in one burr hole lasted for 10 min and kept syringes in CA1 for 5 min after injection to make sure there was sufficient infusion of Aβ_1-42_ oligomers. The incision was sutured and wiped with iodophor. An intramuscular injection of penicillin (40,000 U) should keep for three days for postoperative care [38]. Among all experimental rats, two rats in the sham control group died when operating the intrahippocampal injections. There were no further deaths during this experiment.

### 4.5. Morris Water Maze Test

The Morris water maze (MWM) test is an effective examination to access the spatial learning and memory capability [43]. The MWM test was universally applied in basic research such as neurobiology, pharmacology, and so forth [33]. Shibani et al. applied the MWM test to detect the effects of oleuropein on learning and memory. It was proved that oleuropein could effectively alleviate learning and memory impairment in morphine-treated rats by inhibiting neuronal apoptosis and oxidative stress [44]. The MWM test was also pervasively used to evaluate cognition impairments, which exist in various AD models [45,46,47]. The experimental system mainly contains a water maze device, and an automatic image acquisition and software analysis system [48]. The water maze device consists of a cistern (60 cm in height and 180 cm in diameter) and a movable black platform (12 cm in diameter). The cistern was divided into four quadrants and the platform was placed into the center of a quadrant. The temperature of water was maintained at 24 ± 1 °C and the altitude was 2 cm above the platform. The experiment was completed in an enclosed quiet and gloomy space to reduce the influence of light and noise on learning and memory of experimental rats.

The water maze was carried out on 11th day after microinjection. According to the method of Tian et al. and Staay et al., the test includes a place navigation test and a spatial probe test [49,50]. At the first four days, each rat was trained three times a day in three different quadrants (except the quadrant which the platform was located) in a fixed order and was allowed at least 15 min to rest for a physical recovery between trails. The maximum of the escape latency was 90 s and the rat was allowed to stay at the platform for 15 s before leaving the cistern. The rat was guided to the platform and remained in there for 15 s if it failed to find the platform within 90 s. The escape latency was recorded for each trial. In a spatial probe test, the platform was removed and the rat was allowed to swim for 60 s in the water. The probe latency (the time of rats first reach the platform) and the search frequency (the number of times that rats crossed platform) were recorded respectively.

### 4.6. Passive Avoidance Test

The passive avoidance test was operated according to the method of Shiga et al. [51] and Ji et al. [52]. The experimental equipment was separated into two chambers by an auto-guillotine door, the bright one and the dark one. The test included the training and the testing phase. On the first day of the training phase, all rats were placed into the bright chamber with the door opened for 3 minutes so that the rats can cross-over into the dark chamber instinctively. The next day, all rats were placed in the bright chamber with the auto-guillotine door opened. When the rats went into the black chamber with all four paws, the door was closed and the rats were given an electric shock (0.3 mA, 3 s) [51]. For the testing phase, the rats were placed in the bright chamber again and the response latency to enter into the dark chamber within 3 min was recorded. The latency was recorded as 180 s if the rat was not entering into the dark chamber [52].

### 4.7. Sample Collection and Preparation

Tan IIA was suspended in 0.5% sodium carboxymethyl cellulose (CMC-Na) solution, and the rats of the sham control group and the model group were orally administered with a dose of 100 mg/kg weight. The control group was orally administered with equivalent 0.5% CMC-Na solution. There was fasting and water-drinking free for 12 h before gavage.

#### 4.7.1. Plasma Sample Collection

After being orally administered, all rats were placed in metabolism cages. The blood samples approximately 0.5 mL each time were taken from the suborbital venous plexus at 0.5, 1, 1.5, 2, 4, 6 and 8 h after gavage. All blood samples were collected in heparin sodium anticoagulant EP tubes and rested for 15 min followed by centrifuging at 3000 rpm (4 °C) to separate plasma. The blood supernatants of same timing blood samples were absorbed and gathered into a collective one to obtain the test plasma (sham control and model groups) and blank plasma (control group).

#### 4.7.2. Urine and Feces Sample Collection

Urine samples (0–24 h) were collected by metabolism cages and centrifuged at 3000 rpm (4 °C) for 15 min. The supernatants from the same group were absorbed and merged into a collective one to obtain test samples and blank samples. Feces samples (0–24 h) were collected by metabolism cages and gathered into a collective one. The collective sample was milled and mixed followed by drying. The dried samples (1 g) were homogenized with physiological saline (2 mL) and ultrasonic extracted with ethyl acetate (1 mL) for three times. Ethyl acetate was gathered, centrifuged at 3000 rpm (4 °C) for 15 min and the supernatants were absorbed to obtain test samples and blank samples.

#### 4.7.3. Tissue Sample Collection

After the plasma, urine and faeces sample collection, 7 rats from each group were sacrificed. Brain, heart, liver and kidney were separated and washed with pre-cooled saline. 0.1 g tissue was obtained from every brain and gathered into a collective sample (0.7 g). The collective sample was homogenized with physiological saline (1 mL) and ultrasonic extracted with ethyl acetate (1 mL) three times. Ethyl acetate from every ultrasonic extraction was gathered and centrifuged at 3000 rpm (4 °C) for 15 min. In sequence, the supernatants were acquired to obtain test brain samples and blank samples. With the same process as the brain samples, heart, liver and kidney samples were prepared. The remaining animals in the sham control and model groups were anesthetized by sodium pentobarbital (70 mg/kg, i.p.) and cardiac perfused with physiological saline and 4% paraformaldehyde in sequence. The brains of cardiac perfused rats were taken out and fixed in 4% paraformaldehyde at 4 °C overnight. Fixed brain tissues were embedded in the paraffin and sliced with a thickness of 5 μm for further Nissl staining.

#### 4.7.4. Sample Preparation

The SPE cartridge was used to pretreat samples for concentration and precipitation of protein and solid residue. At first, the SPE cartridges were activated with methanol (5 mL) and deionized water (5 mL) in sequence. Then, plasma (1 mL), urine (1 mL) and tissue (2.5 mL) samples were respectively added into the SPE cartridge. The cartridges were washed by deionized water (5 mL) and methanol (3 mL) successively. The methanol eluate was collected into an EP cube and dried via nitrogen at room temperature. Finally, the residue was re-dissolved in 10% acetonitrile (80 μL) and centrifuged at 13,500 rpm (4 °C) for 30 min. The supernatant was collected for metabolic analysis. 

### 4.8. Nissl Staining

Brain tissue slices, which with a thickness of 5 μm, were dewaxed in xylene three times (5 min for each time). Then, the slices were washed successively by ethanol of 100% (5 min), 90% (2 min) and 70% (2 min) followed by distilled water (2 min). The slices were soaked with Nissl stain for 10 min at 37 °C and washed with distilled water two times (2 s for each time) and 95% ethanol for 5 s. At last, the slices were dehydrated with 95% ethanol two times (2 min for each time), and transparentized with xylene for 5 min. Following this, the slices were transparentized with fresh xylene for another 5 min and sealed with neutral resin for observation. The images were captured with a microscope (Nikon, Japan) and quantified by ImageJ, version 1.50 (National Institutes of Health, Bethesda, MD, USA).

### 4.9. Instruments and Analytical Conditions

The UHPLC-Q-Exactive mass spectrometer (Thermo Scientific, Bremen, Germany) was ultilized to identify the metabolites of tan IIA. Chromatographic separation was operated on a Waters ACQUITY UHPLC^®^ BEH C18 column (2.1 mm × 50mm i.d., 1.7 μm; Waters Corporation, Milford, MA, USA). A linear gradient elution of mobile phase consisted of water containing 0.1% formic acid (A) and acetonitrile (B) which was applied with the following program: 0–2 min, 5–20% B; 2–25 min, 20–85% B; 25–30 min, 5% B. The temperature of the column and the auto sampler were set at 30 °C and 10 °C respectively. The flow rate was 0.4 mL/min and the injection volume of all samples was 2 μL. 

The mass spectrometer was processed in the positive ion mode and the parameters were set as follows: Spray voltage, 4.0 kV; capillary temperature, 350 °C; capillary voltage of 30 V sheath gas, 40 arb; auxiliary gas, 20 arb; probe heater temperature, 300 °C. A high-resolution MS analysis was operated at full scan with a mass range of *m*/*z* 200–1500 at a resolving power of 70,000 (full width at half maximum FWHM as defined at *m*/*z* 400), and at the MS^2^ data with a resolution of 17,500 triggered by data-dependent scanning. Helium served as the collision gas. Collision-induced dissociation was carried out with an isolation width of 2.0 Da, and 30% maximum of collision energy.

### 4.10. Data Processing and Statistical Analyses

The collected data was processed and analyzed via Thermo Xcalibur 2.1 workstation (Thermo Scientific, Bremen, Germany). The peaks were picked when their intensity were over 10000 under the positive ion mode so that more fragments could be acquired. Regarding accurate mass, possible element composition, and possible reaction, the parameters of the formula predictor were set as follows: C [0–30], H [0–60], O [0–15], S [0–3], N [0–3] and ring double bond (RDB) equivalent value [0–15]. The maximal errors between theoretical mass and experimental mass were limited within 4 ppm. 

All results were analyzed via SPSS software package, version 19.0 (SPSS, Inc., Chicago, IL, USA). The difference between groups was analyzed by one-way analysis of variance (ANOVA) followed by a Tukey test for multiple comparisons. It is stipulated that *p* < 0.05 indicates it is statistically significant.

## 5. Conclusions

UHPLC-Q-Exactive Orbitrap mass spectrometry, combined with the MWM test, the passive avoidance test and Nissl staining was used to identify the metabolites of tan IIA in a Aβ_1–42_ induced AD rat model. By analyzing the metabolite information, including characteristic fragments in the ESI-MS/MS spectra, retention time and other relative data, a total of 37 metabolites were identified under a positive ion mode. The proposed metabolic pathway of tan IIA involved dehydrogenation, hydroxylation, demethylation, hydrolysis, decarbonylation and combinations with glucuronide and glycine. It is noted that **M12** was detected in brain sample of the sham control group and **M20, M27** were only identified in the Aβ_1–42_ induced AD model group rats. Tan IIA exerts significant effects on protecting neurons and alleviating AD, and our study provided indispensable information for research on metabolites of tan IIA in vivo and on probable active structures of tan IIA to exert neuroprotection, such as **M20** and **M27**.

## Figures and Tables

**Figure 1 molecules-24-02584-f001:**
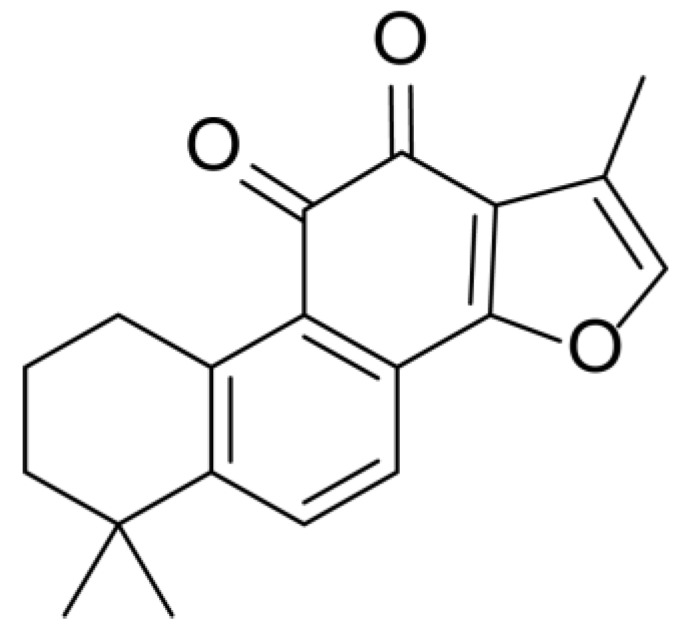
Structure of tanshinone IIA.

**Figure 2 molecules-24-02584-f002:**
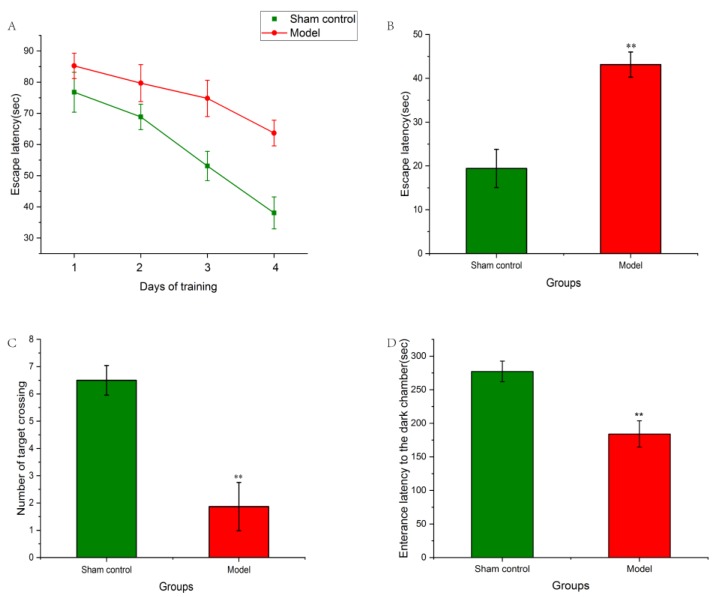
The effects of Aβ_1–42_ on a Morris water maze test and passive avoidance test. (**A**) Average escape latency of two groups onto a hidden platform. Three training trials were administered each day for four continuous days. (**B**) Spatial probe test. Escape latency was defined as the time the rats first crossed the point of the platform. (**C**) Search frequency was defined as the number of times that rats crossed the point of the platform. (**D**) The effect of Aβ_1–42_ on passive avoidance test. ** *p* < 0.01 when compared with sham control group.

**Figure 3 molecules-24-02584-f003:**
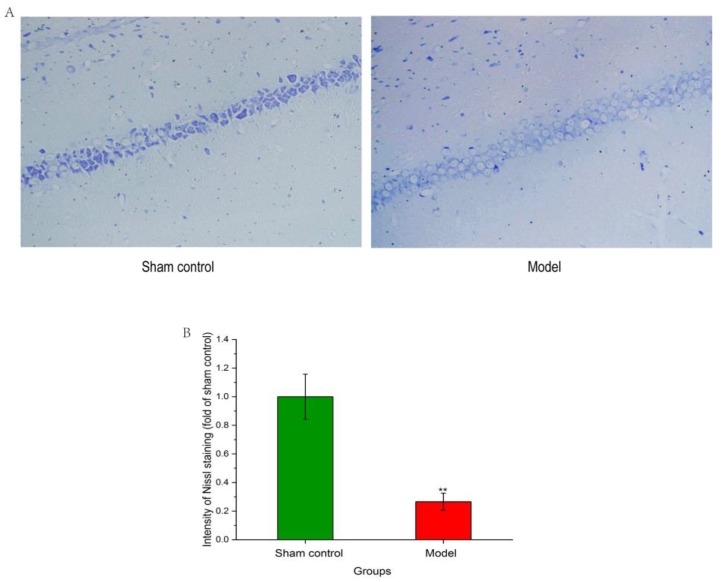
The effects of Aβ_1–42_ on neuronal structure in the hippocampus tissues. (**A**) Nissl staining in hippocampus of the sham control and the model group. (**B**) Quantitative analysis of Nissl staining intensity. ** p < 0.01 when compared with sham control group.

**Figure 4 molecules-24-02584-f004:**
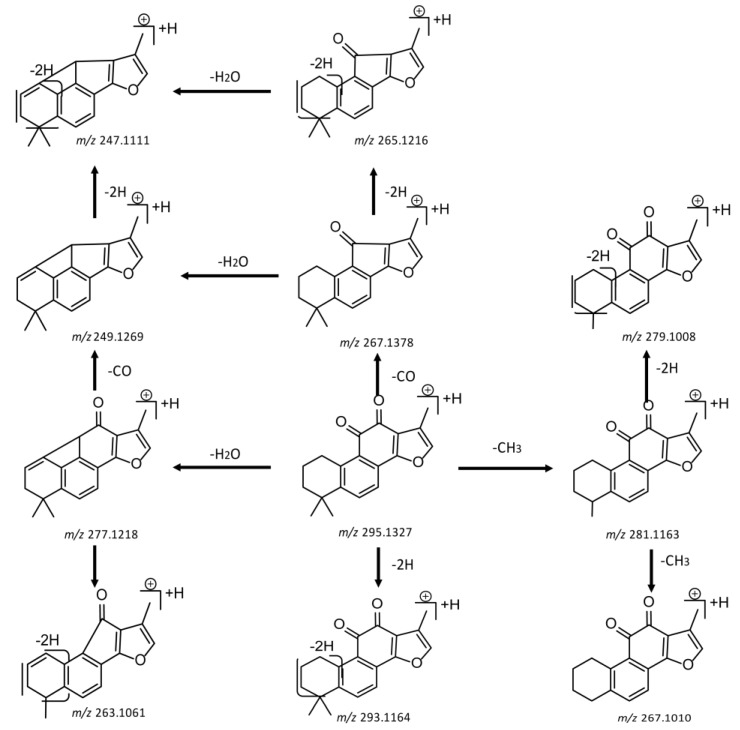
The fragment behaviors of tanshinone IIA in a positive ion mode.

**Figure 5 molecules-24-02584-f005:**
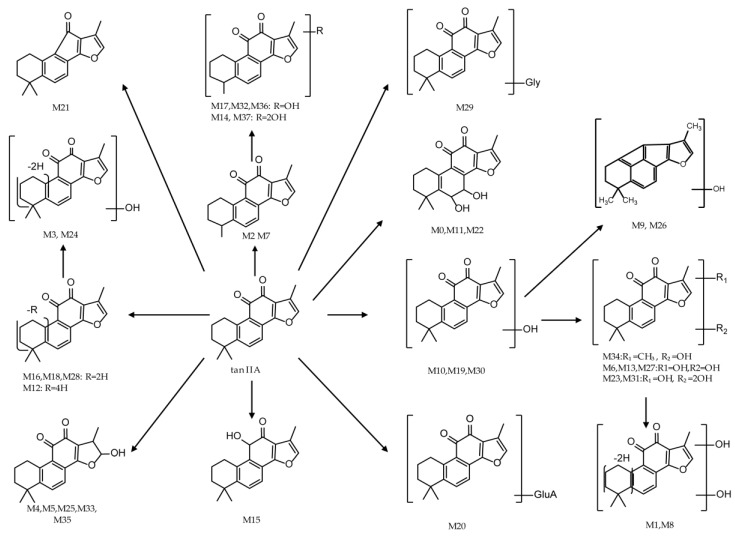
The proposed tanshinone IIA metabolic patterns in vivo.

**Figure 6 molecules-24-02584-f006:**
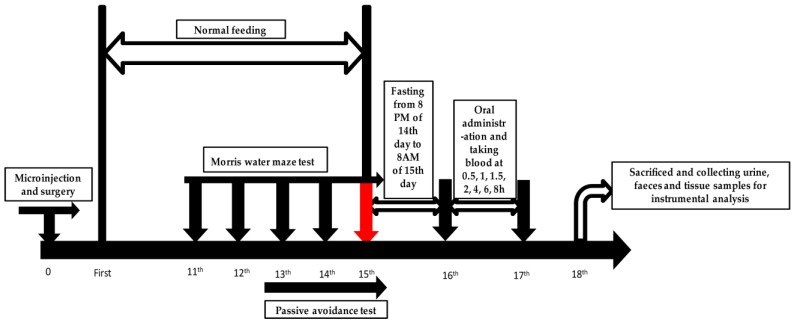
Experimental process of AD model establishment and metabolic analysis.

**Table 1 molecules-24-02584-t001:** Summary of tanshinone IIA metabolites by HPLC-Q-Exactive orbitrap.

Peak	t_R_/min	Formula [M + H]^−^	Theoretical Mass *m*/*z*	Experimental Mass *m*/*z*	Error (ppm)	MS/MS Fragment Ions	Identification/Reactions	Distribution	Sham Group	Model Group
**M0**	4.25	C_19_H_21_O_5_	329.1383	329.13809	−0.791	MS^2^[329]:283.1321(100),311.1268(50),265.1219(15),267.1371(11)	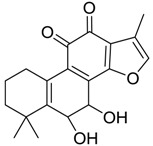	2	+	+
**M1**	4.36	C_19_H_17_O_5_	325.107	325.10681	−0.739	MS^2^[325]:156.0764(100),110.0715(59),307.0952(26),279.1008(23),168.1015(22)	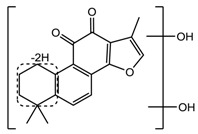	1	+	+
**M2**	4.75	C_18_H_17_O_3_	281.1172	281.11703	−0.679	MS^2^[281];133.0645(100),263.1061(18),282.18024(12),281.1166(10)	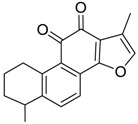	1	+	
**M3**	4.81	C_19_H_17_O_4_	309.1121	309.11163	−3.409	MS^2^[309]:204.1378(100),267.1695(12)	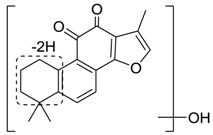	1	+	
**M4**	5.71	C_19_H_21_O_4_	313.1435	313.14383	1.259	MS^2^[313]:85.0289(100),313.2163(18),267.1378(13),295.2051(12)	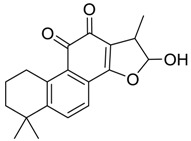	1		+
**M5**	5.72	C_19_H_21_O_4_	313.1435	313.14368	0.78	MS^2^[313]:313.2150(100),159.1167(54), 295.2049(53.32), 277.1943(35),314.2175(26)	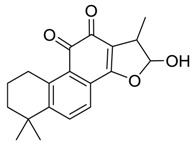	7	+	
**M6**	6.2	C_19_H_19_O_5_	327.1227	327.1226	−0.062	MS^2^[327]:327.1219(100),309.1113(69),281.1166(61),257.1166(40),328.2262(26)	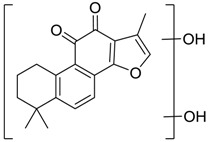	1,2	+	+
**M7**	6.22	C_18_H_17_O_3_	281.1172	281.11716	−0.217	MS^2^[281]:255.1009(100),149.0230(72),224.2004(47),263.1636(42),207.1012(36),279.1578(36),95.0860(35),81.0705(31)	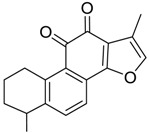	1	+	+
**M8**	6.6	C_19_H_17_O_5_	325.107	325.10724	0.584	MS^2^[325]:325.1060(100),281.1164(21)	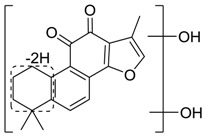	2	+	+
**M9**	6.84	C_18_H_17_O_2_	265.1223	265.12241	0.391	MS^2^[265]:247.1685(100),187.1476(51),191.1059(27),229.1576(23)	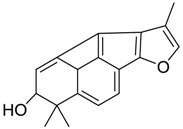	1	+	+
**M10**	6.85	C_19_H_19_O_4_	311.1278	311.12766	−0.404	MS^2^[311]:265.1216(100),283.1315(46),266.1237(20),223.0749(15),311.1257(13),223.1116(10)	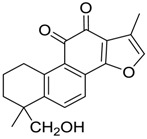	1	+	
**M11**	7.45	C_19_H_21_O_5_	329.1383	329.1385	0.607	MS^2^[329]:85.0653(100),311.2002(57),330.2246(41),109.1014(44)	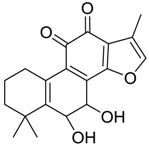	1	+	+
**M12**	8.33	C_19_H_15_O_3_	291.1015	291.10138	−2.539	MS^2^[291]:290.1094(100),292.1068(43),249.1576(24),273.1848(14)	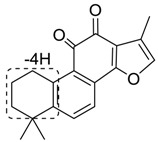	1,2,3,4,5,6	+	
**M13**	8.7	C_19_H_19_O_5_	327.1227	327.12234	−2.777	MS^2^[327]:281.1163(100),309.1108(99),327.1212(96),273.1114(40),256.1088(29),268.1200(18)	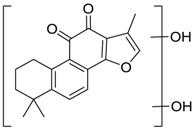	1	+	+
**M14**	8.75	C_18_H_17_O_5_	313.107	313.10617	1.477	MS^2^[313]:285.1116(100),313.1061(63),267.1010(37),295.0950(22),163.1107(13)	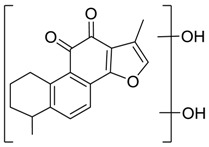	2	+	+
**M15**	9.01	C_19_H_21_O_3_	297.1486	297.14819	−1.114	MS^2^[297]:297.1117(100),279.1011(18),251.1056(13),269.1168(11)	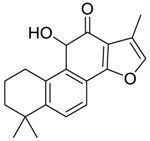	2	+	
**M16**	9.13	C_19_H_17_O_3_	293.1172	293.11688	−1.163	MS^2^[293]:293.11655(100),275.1055(87),247.1115(37)	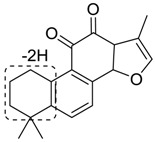	1	+	+
**M17**	9.14	C_18_H_17_O_4_	297.1121	297.11234	0.688	MS^2^[297]:297.1117(100),95.0859(27),226.6350(20),251.1056(13),269.1168(10)	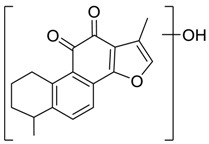	2	+	
**M18**	9.27	C_19_H_17_O_3_	293.1172	293.11694	−2.829	MS^2^[293]:293.1160(100),226.89243(51),275.1056(32)	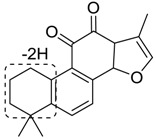	1	+	+
**M19**	9.27	C_19_H_19_O_4_	311.1278	311.1275	−0.918	MS^2^[311]:275.1058(100),293.1164(72),196.1691(40),311.1268(39),265.1217(33),283.1163(12),	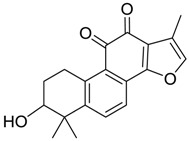	1,2,3,4,5	+	+
**M20**	9.46	C_25_H_27_O_9_	471.165	471.16409	−1.844	MS^2^[471]:295.1321(100),277.1219(19)	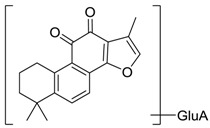	1		+
**M21**	9.6	C_18_H_19_O_2_	267.138	267.13748	−1.783	MS^2^[267]:267.1369(100),225.1957(42),238.9165(18),211.06177(14)	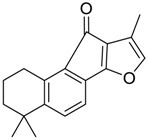	2	+	+
**M22**	9.61	C_19_H_21_O_5_	329.1383	329.13785	−1.52	MS^2^[329]:329.1373(100),299.0907(91),311.1269(39),283.1310(12),109.1012(12)	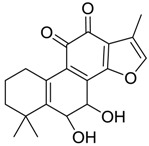	1	+	+
**M23**	9.61	C_19_H_19_O_6_	343.1176	343.11746	−0.451	MS^2^[343]:279.1009(100),283.0957(63),325.1058(46),343.1167(31),297.1111(21),253.0853(17)	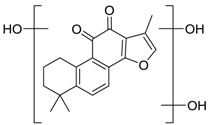	1	+	
**M24**	9.76	C_19_H_17_O_4_	309.1121	309.1116	−3.507	MS^2^[309]:265.1217(100),309.1113(82)	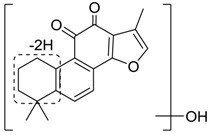	2	+	
**M25**	9.87	C_19_H_21_O_4_	313.1435	313.14352	0.269	MS^2^[313]:84.9603(100),267.9286(58),313.1438(34),219.9089(14),238.9163(10)	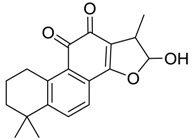	1,2	+	+
**M26**	10.28	C_18_H_17_O_2_	265.1223	265.12225	−0.212	MS^2^[265]:265.1215(100),201.0485(30),210.0539(20)	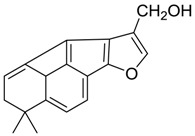	2	+	
**M27**	10.3	C_19_H_19_O_5_	327.1227	327.12253	−0.52	MS^2^[327]:281.1165(100),309.1111(32),257.1163(14)	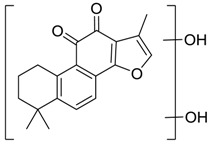	2		+
**M28**	10.62	C_19_H_17_O_3_	293.1172	293.11682	−3.239	MS^2^[293]:293.1165(100),275.1059(31),247.1111(25),278.0929(13)	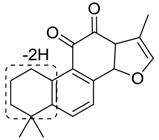	1	+	+
**M29**	10.7	C_21_H_22_O_4_N	352.1544	352.15411	−0.638	MS^2^[352]:333.1089(100),352.1579(39),310.1425(38),210.1493(12),283.1340(10)	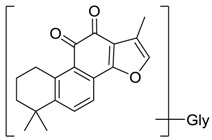	1,2,5	+	
**M30**	11.28	C_19_H_19_O_4_	311.1278	311.12796	0.561	MS^2^[311]:293.1163(100),311.1261(39),275.1052(25),247.1188(24),265.1222(17),286.6728(15),283.1318(10)	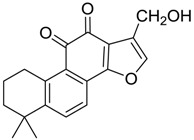	2	+	
**M31**	11.48	C_19_H_19_O_6_	343.1176	343.11761	-0.014	MS^2^[343]:323.2511(100),341.2621(56),325.1058(44),343.1141(26),297.1100(20)	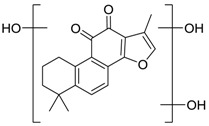	1	+	
**M32**	11.58	C_18_H_17_O_4_	297.1121	297.11255	1.395	MS^2^[297]:297.1112(100),81.0704(24),279.1004(19),97.1015(11)	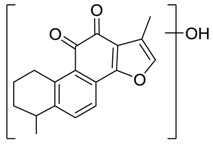	2	+	+
**M33**	11.93	C_19_H_21_O_4_	313.1435	313.14343	−0.018	MS^2^[313]:313.1421(100),271.1320(32),297.1496(28),267.1379(25),295.1332(17),107.0859(11)	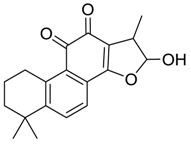	2	+	+
**M34**	12.37	C_20_H_21_O_4_	325.1434	325.14365	0.659	MS^2^[325]:325.1423(100),265.1214(43),209.1167(41),279.1377(18)	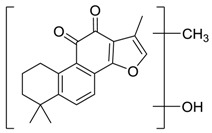	2	+	
**M35**	13.09	C_19_H_21_O_4_	313.1435	313.14322	−0.689	MS^2^[313]:313.1419(100),295.1322(39),267.1371(18),83.96021(11)	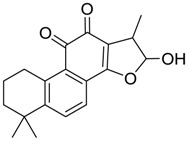	2	+	+
**M36**	13.53	C_18_H_17_O_4_	297.1121	297.11194	−2.504	MS^2^[297]:269.1164(100),287.1276(21),297.1116(20),270.1204(19),109.1012(12)	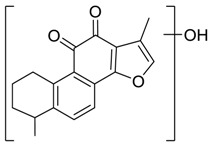	1	+	+
**M37**	13.68	C_18_H_17_O_5_	313.107	313.10715	0.319	MS^2^[313]:285.1117(100),269.1169(85),313.1065(58),267.10065(41)295.0972(26)	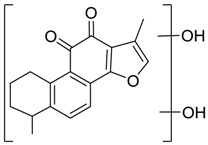	2	+	+

Note: t_R_: retention time; 1: urine; 2: feces; 3: heart; 4: liver; 5: kidney; 6: brain; 7: plasma; +: detected.

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
