# Peer review of "Rapid Identification of Tanshinone IIA Metabolites in an Amyloid-β1-42 Induced Alzherimer’s Disease Rat Model using UHPLC-Q-Exactive Qrbitrap Mass Spectrometry"

_molecules, 2019, doi:10.3390/molecules24142584_

Round 1

Reviewer 1 Report

Liang et. al. described the metabolites in relation to tanshinone IIA. The results were straightforward and convincing. However, there are a few improvements that should be made before publication.

Major

1.     In the Morris water maze test, AD model rats showed the impairment of spatial reference memory, compared to the sham group. Assessment of motor function is needed in order to accurately evaluate cognitive function. Total swimming distance or velocity in the experiment would be useful information.

2.     In the evaluation of neuronal structure in the hippocampus, Nissl staining was performed. However, quantitative analysis was not performed. The authors should add it.

3.     Morris water maze test gives great stress to the mice. Body weight is usually decreased through the tests. The reviewer is wondering whether the amount of food intake and body weight was different between the AD model and sham rats. If so, the difference would affect the metabolism of tanshinone IIA. The authors should clarify the amount of food intake and the changes of the body weight during the experiment. If the authors do not have the relevant data, the metabolites of tanshinone IIA detected in the current study should be confirmed in a replication study. In the replication study, the behavioral experiments should not be performed.  

4.     The levels of several metabolites were different between the AD model and sham rats (page17 line 256 - 265). However, the description is not enough. The authors should describe the detail about the differences.

Minor

1.     Twenty-five rats were purchased for this experiment (page 18, line 285). However, the amount of seven blank biological samples, ten sham control, and ten AD model is twenty-seven. Please clarify it.

2.     Perioperative mortality rates should be described.

3.     The authors should describe the title of the current animal experiment which was approved by the ethical committee, the date of approval, and the authorization number.

4.     The discussion is not enough. The authors should explain the significance of the results in the discussion.

Author Response

Major

1.     In the Morris water maze test, AD model rats showed the impairment of spatial reference memory, compared to the sham group. Assessment of motor function is needed in order to accurately evaluate cognitive function. Total swimming distance or velocity in the experiment would be useful information.

Response: Thank you for your suggestion. But it is regretful that total swimming distance and velocity of the Morris water maze test were not recorded. In future experiments, we’ll also focus on the changes of these indexes. 

2.     In the evaluation of neuronal structure in the hippocampus, Nissl staining was performed. However, quantitative analysis was not performed. The authors should add it.

Response: Thank you for your suggestion. Quantitative analysis has been added in the part of “2.3 Effect of Aβ1-42 on Hippocampal Neuronal Structure of AD Rat Model”.

3.     Morris water maze test gives great stress to the mice. Body weight is usually decreased through the tests. The reviewer is wondering whether the amount of food intake and body weight was different between the AD model and sham rats. If so, the difference would affect the metabolism of tanshinone IIA. The authors should clarify the amount of food intake and the changes of the body weight during the experiment. If the authors do not have the relevant data, the metabolites of tanshinone IIA detected in the current study should be confirmed in a replication study. In the replication study, the behavioral experiments should not be performed.  

ResponseThank you for your question. It is fortunate that the amounts of food intake and body weight were recorded respectively during the experiment operated. Analyzing the differences between sham control group and model group by SPSS 19, it is concluded that there were no significant differences between the two groups, both for food intake and body weight. Moreover, Morris water test didn’t significantly decreased the weight of experimental rats. And relative descriptions has been complemented in the paper.

4.     The levels of several metabolites were different between the AD model and sham rats (page 17 line 256 - 265). However, the description is not enough. The authors should describe the detail about the differences.

Response The details about distributional differences have been descripted in the manuscript.  

Minor

1.         Twenty-five rats were purchased for this experiment (page 18, line 285). However, the amount of seven blank biological samples, ten sham control, and ten AD model is twenty-seven. Please clarify it.

Response: The number of rats has been corrected to twenty-seven. Thank you for your correction.

2.         Perioperative mortality rates should be described.

Response: The mortality rate of experimental rats was complemented in the part of “4.4 Microinjection and Surgery”.

3.     The authors should describe the title of the current animal experiment which was approved by the ethical committee, the date of approval, and the authorization number.

Response: Relative information have been completed in the part of “4.3 Animals and Drug Administration”.

4.     The discussion is not enough. The authors should explain the significance of the results in the discussion.

Response: Discussion has been complemented in the part of “3. Disscussion”.

Reviewer 2 Report

This publication described the rapid idenification of tan IIA metabolites in Alzheimer's disease rat model using mass spectomety method. The results are new, interesting, and provided valuable references for Research on metabolites of tan IIA n vivo. However, this manuscript need a substantial improvement.

Keywords:

Please also include: hippocampus; rat model; morris water maze test; passive avoidance test

Introduction:

Lines 39-49: Please  include more references.

Line 65: However, few studies reported the detailed metabolites of tan IIA... Please, describe all previosly results of these few previous studies!!!! ....

Results and Discussion:

In all part I see only the results, but absolutely without any discussion?? Please, include in all Sections (from 2.1 until 2.5  not only your results, but also the discussion, Plese discuss all your results!!!

Lines 257-265: Please, compare your findings with the available literature.

Materials and Methods:

Line 287: What about water and food access ?

Line 292: Please, include the Protocol number, Instutution  and Date  - exactly!! 

Line 300: Please, use a new one edition of this Atlas

Line 303: were injected bilaterally with steril water? Because usully it is PBS + BSA.. Why only water?

Line 309: Please include more references

Line 311: Please describe also the depth of this cistern.

Line 316: the WMZ test was carried out on 11th day after microinjection. But it is very early after the bilaterally injection into hippocampus? Usually, you should wait about  4-6 weeks after the intracerebrally injection, especially after bilaterally  injection. What about the animals health?

WMZ test: Which test did you use? According to ... Plese, describe..

WMZ test: Which light condition did you use? Plese, describe..

3.6. Passive avoidance test:

According to..? Please, include  the exactly references?

Line 342: Which apparatus did you use?

Line 343: taken from suborbital venous Plexus.. Without to anesthetized? Which anesthesia did you use?

Line 340: fasting and water-drinking free for 12 h before? Ist it nornal and gut for rats?  

Line 364: were anesthetized.. How? Line 366: Please, include more details..

All part (Nissl staining 3.8. ): please describe more exactly..

Line 380: covered ..? How?

       Conclusions:

Line 418:... ist probably ative sructure on exerting neuroprotection.. Plese, include .. in experimental AD rat model.

References:

Please, correct all references,  they are done not according to "Molecles"!!

Author Response

Keywords:

Please also include: hippocampus; rat model; morris water maze test; passive avoidance test

Response: All the above keywords was added into the revised manuscript.

Introduction:

Lines 39-49: Please include more references.

Response: More references has been added into this part.

Line 65: However, few studies reported the detailed metabolites of tan IIA... Please, describe all previosly results of these few previous studies!!!! ....

Response: Previous studies on the metabolites of tan IIA have been reviewed in the second paragraph of the “Introduction”.

Results and Discussion:

In all part I see only the results, but absolutely without any discussion?? Please, include in all Sections (from 2.1 until 2.5  not only your results, but also the discussion, Plese discuss all your results!!!

Response: Discussion has been complemented in the part of “3. Disscussion”.

Lines 257-265: Please, compare your findings with the available literature.

Response: The comparison has been added in the part of “3. Discussion”.

Materials and Methods:

Line 287: What about water and food access ?

Response: They were allowed to access water and food ad libitum.

Line 292: Please, include the Protocol number, Instutution  and Date  - exactly!! 

Response: Relative information have been completed in the part of “4.3 Animals and Drug Administration”.

Line 300: Please, use a new one edition of this Atlas

Response: A newer Atlas was cited in the manuscript.

Line 303: were injected bilaterally with steril water? Because usully it is PBS + BSA.. Why only water?

Response: Thank you for your question. Firstly, I admitted that I made a mistake when writing the paper. The sham control group were injected with sterile PBS because the PBS was the vehicle of aggregated soluble Aβ1-42 oligomers.

Line 309: Please include more references

Response: Some references have been added in the part of “4.5 Morris Water Maze Test”.

Line 311: Please describe also the depth of this cistern.

Response: The depth of this cistern is 60 cm. Relative descriptions are as following: The water maze device consists of a cistern (60 cm in height and 180 cm in diameter) and a movable black platform (12 cm in diameter)

Line 316: the WMZ test was carried out on 11th day after microinjection. But it is very early after the bilaterally injection into hippocampus? Usually, you should wait about  4-6 weeks after the intracerebrally injection, especially after bilaterally  injection. What about the animals health?

Response: On the 11th day after microinjection, the rats were vibrant and able to take food and water normally. Besides, the incision on the head of experimental rats recovered well and only a shallow trace could be seen. Overall, the condition of rats were capable for behavior experiments, including the MWM test and passive avoidance test.

WMZ test: Which test did you use? According to ... Plese, describe..

Response: Relative description has been added in the part of “4.5 Morris Water Maze Test”.

WMZ test: Which light condition did you use? Plese, describe..

Response: The light condition during Morris Water Maze was described in the part of “4.5 Morris Water Maze Test”.

3.6. Passive avoidance test:

According to..? Please, include  the exactly references?

Response: The exactly references have been included in the part of “Passive avoidance test”.

Line 342: Which apparatus did you use?

Response: Metabolism cages were used to collect urine and faeces.

Line 343: taken from suborbital venous Plexus.. Without to anesthetized? Which anesthesia did you use?

Response: In this experiment, urine and faeces were collected for metabolic analysis of tan IIA. We were afraid that anesthetizing could influence the metabolic pathway of tan IIA. In addition, anesthetizing could limit food intake and drink of experimental rats, which could decrease the production of urine and faeces. In case of the happening of above situation, the rats were taken bloods without to anesthetized.

Line 340: fasting and water-drinking free for 12 h before? Ist it nornal and gut for rats?  

Response: Fasting and water-drinking free for 12h or overnight is normal for rats. On one hand, fasting can accelerate the absorption of medicals. On the other hand, the food residues in the stomach of rats might affect the metabolic pathways of tan IIA. So fasting was operated for 12h before gavage. 

eg.

[1] Luo, H.; Jiang, M.; Lian, G.; Liu, Q.; Shi, M.; Li, T.Y.; Song, L.; Ye, J.; He, Y.; Yao, L., et al. AIDA Selectively Mediates Downregulation of Fat Synthesis Enzymes by ERAD to Retard Intestinal Fat Absorption and Prevent Obesity. Cell Metab 2018, 27, 843-853.e846, doi:10.1016/jNaNet.2018.02.021.

[2] Traesel, G.K.; Menegati, S.E.; Dos Santos, A.C.; Carvalho Souza, R.I.; Villas Boas, G.R.; Justi, P.N.; Kassuya, C.A.; Sanjinez Argandona, E.J.; Oesterreich, S.A. Oral acute and subchronic toxicity studies of the oil extracted from pequi (Caryocar brasiliense, Camb.) pulp in rats. Food Chem Toxicol 2016, 97, 224-231, doi:10.1016/j.fct.2016.09.018.

Line 364: were anesthetized.. How?

Response: The rats were anesthetized by intraperitoneal injection with sodium pentobarbital (70 mg/kg, i.p.).

Line 366: Please, include more details..

Response: Some details have been complemented in the part of “4.7.3 Tissue Sample Collection”.

All part (Nissl staining 3.8. ): please describe more exactly..

Line 380: covered ..? How?

Response: Relative details have been added in the part of “4.8 Nissl staining”.

Conclusions:

Line 418:... ist probably ative sructure on exerting neuroprotection.. Plese, include .. in experimental AD rat model.

Response: Possible active structures of tan IIA includes M20 and M27. And relative descriptions have been added in the part of Conclusions.

References:

Please, correct all references,  they are done not according to "Molecles"!!

Response: Thank you for your correction. All references have been cited as the styles of “Molecules”.

Round 2

Reviewer 1 Report

The authors properly clarified the questions that had been pointed out.

Author Response

Thank you for your review 

Best wishes for you 

Reviewer 2 Report

Thank you very much.    

Unfortunately, before publishing, this manuscript need once more a few corrections:

Line 35: correct tan IIA please.

Line 68: please correct 3-kinase/Akt

Line 132: please correct a multiple dots.

Line 307 and 353: please correct ....[24] and ..[42].

Is it possible to improve the quality of all Figures? Please correct all Legends according to IJMS. But, very important for me to clarify one again the question: Line 404-405: I think, according to literature and my own experience and taken care of the experimental animals, it is not possible to take a blood samples from suborbital venous plexus without anesthesia. Did you do this manipulation without anesthesia? Please see one example from many others: 

J Pharmacol Pharmacother. 2010 Jul-Dec; 1(2): 87–93.

doi: 10.4103/0976-500X.72350

PMCID: PMC3043327

PMID: 21350616

Blood sample collection in small laboratory animals

S Parasuraman, R Raveendran, and R Kesavan

S Parasuraman

Department of the Pharmacology, Jawaharlal Institute of Postgraduate Medical Education and Research, Pondicherry, India

Find articles by S Parasuraman

R Raveendran

Department of the Pharmacology, Jawaharlal Institute of Postgraduate Medical Education and Research, Pondicherry, India

Find articles by R Raveendran

R Kesavan

Department of the Pharmacology, Jawaharlal Institute of Postgraduate Medical Education and Research, Pondicherry, India

Find articles by R Kesavan

Author information Copyright and License information Disclaimer

Department of the Pharmacology, Jawaharlal Institute of Postgraduate Medical Education and Research, Pondicherry, India

Address for correspondence: Parasuraman S, Department of the Pharmacology, Jawaharlal Institute of Postgraduate Medical Education and Research, Pondicherry, India. E-mail: moc.liamg@dhpusarap

Copyright © Journal of Pharmacology and Pharmacotherapeutics

This is an open-access article distributed under the terms of the Creative Commons Attribution License, which permits unrestricted use, distribution, and reproduction in any medium, provided the original work is properly cited.

This article has been corrected. See J Pharmacol Pharmacother. 2017; 8(3): 153.

Author Response

Line 35: correct tan IIA please.

Response: “ tanIIA” has been corrected as “tan IIA”

Line 68: please correct 3-kinase/Akt

Response: “3-kinase/ Akt” has been corrected as “3-kinase/Akt”

Line 132: please correct a multiple dots.

Response: Multiple dots of Line 115,117,118,131 and 132 has been deleted.

Line 307 and 353: please correct ....[24] and ..[42].

Response: The reference [24] has been corrected as

Lin, L.; Jadoon, S.S.; Liu, S.Z.; Zhang, R.Y.; Li, F.; Zhang, M.Y.; Ai-Hua, T.; You, Q.Y.; Wang, P. Tanshinone IIA ameliorates spatial learning and memory deficits by inhibiting the activity of ERK and GSK-3beta. J Geriatr Psychiatry Neurol 2019, 32, 152-163, doi:10.1177/0891988719837373”

The reference [42] has been corrected as

“Paxinos, G.; Watson, C. The Rat Brain in Stereotaxic Coordinates, 7th edition; Academic Press: Cambridge, Massachusetts, 2013; pp. 1-472, ISBN 978-0-12-391949-6..”

Is it possible to improve the quality of all Figures? Please correct all Legends according to IJMS. 

Response: All figures, supplied in the folder of “Fig and Tab. zip”, were with a resolution of 500 dpi, which were accordance with the standard of IJMS( a resolution of 300 dpi or higher).

But, very important for me to clarify one again the question: Line 404-405: I think, according to literature and my own experience and taken care of the experimental animals, it is not possible to take a blood samples from suborbital venous plexus without anesthesia. Did you do this manipulation without anesthesia? Please see one example from many others: 

J Pharmacol Pharmacother. 2010 Jul-Dec; 1(2): 87–93.

doi: 10.4103/0976-500X.72350

PMCID: PMC3043327

PMID: 21350616

Blood sample collection in small laboratory animals

S Parasuraman, R Raveendran, and R Kesavan

Author information Copyright and License information Disclaimer

This article has been corrected. See J Pharmacol Pharmacother. 2017; 8(3): 153.

Response: Thank you for your question. We have read the paper “Blood sample collection in small laboratory animals” and other relative papers. It is concluded that taking blood from suborbital venous plexus should be operated with anesthesia. In our experiment, for fear that anesthesia could result in samples insufficient and changes of tan IIA metabolites, we indeed took blood examples from suborbital venous plexus without anesthesia. And I think that the more reliable method is to operate a preliminary experiment to explore the effects of anesthesia(sodium pentobarbital) on metabolic analysis. In our future projects, the experiments to explore the effects of anesthesia on metabolic analysis will be carried on earlier to make sure the veracity of metabolic data and take care of experimental animals. Thank you for your correction once again.
